# Combination Effect of Novel Bimetallic Ag-Ni Nanoparticles with Fluconazole against *Candida albicans*

**DOI:** 10.3390/jof8070733

**Published:** 2022-07-14

**Authors:** Majid Rasool Kamli, Elham A. Alzahrani, Soha M. Albukhari, Aijaz Ahmad, Jamal S. M. Sabir, Maqsood Ahmad Malik

**Affiliations:** 1Department of Biological Sciences, Faculty of Sciences, King Abdulaziz University, P.O. Box 80203, Jeddah 21589, Saudi Arabia; mkamli@kau.edu.sa (M.R.K.); jsabir2622@gmail.com (J.S.M.S.); 2Center of Excellence in Bionanoscience Research, King Abdulaziz University, P.O. Box 80203, Jeddah 21589, Saudi Arabia; 3Chemistry Department, Faculty of Sciences, King Abdulaziz University, P.O. Box 80203, Jeddah 21589, Saudi Arabia; ealzahrani0229@stu.kau.edu.sa (E.A.A.); salbukhari@kau.edu.sa (S.M.A.); 4Clinical Microbiology and Infectious Diseases, School of Pathology, Faculty of Health Sciences, University of the Witwatersrand, Johannesburg 2193, South Africa; aijaz.ahmad@wits.ac.za; 5Infection Control Unit, Charlotte Maxeke Johannesburg Academic Hospital, National Health Laboratory Service, Johannesburg 2193, South Africa

**Keywords:** *Candida albicans*, fuconazole resistance, synergistic effect, efflux pumps, biofilms

## Abstract

The increasing frequency of antifungal drug resistance among pathogenic yeast “*Candida*” has posed an immense global threat to the public healthcare sector. The most notable species of *Candida* causing most fungal infections is *Candida albicans.* Furthermore, recent research has revealed that transition and noble metal combinations can have synergistic antimicrobial effects. Therefore, a one-pot seedless biogenic synthesis of Ag-Ni bimetallic nanoparticles (Ag-Ni NPs) using *Salvia officinalis* aqueous leaf extract is described. Various techniques, such as UV–vis, FTIR, XRD, SEM, EDX, and TGA, were used to validate the production of Ag-Ni NPs. The antifungal susceptibility of Ag-Ni NPs alone and in combination with fluconazole (FLZ) was tested against FLZ-resistant *C. albicans* isolate. Furthermore, the impacts of these NPs on membrane integrity, drug efflux pumps, and biofilms formation were evaluated. The MIC (1.56 μg/mL) and MFC (3.12 μg/mL) results indicated potent antifungal activity of Ag-Ni NPs against FLZ-resistant *C. albicans*. Upon combination, synergistic interaction was observed between Ag-Ni NPs and FLZ against *C. albicans* 5112 with a fractional inhibitory concentration index (FICI) value of 0.31. In-depth studies revealed that Ag-Ni NPs at higher concentrations (3.12 μg/mL) have anti-biofilm properties and disrupt membrane integrity, as demonstrated by scanning electron microscopy results. In comparison, morphological transition was halted at lower concentrations (0.78 μg/mL). From the results of efflux pump assay using rhodamine 6G (R6G), it was evident that Ag-Ni NPs blocks the efflux pumps in the FLZ-resistant *C. albicans* 5112. Targeting biofilms and efflux pumps using novel drugs will be an alternate approach for combatting the threat of multi-drug resistant (MDR) stains of *C. albicans*. Therefore, this study supports the usage of Ag-Ni NPs to avert infections caused by drug resistant strains of *C. albicans*.

## 1. Introduction

Nanotechnology is a safe, green, and clean technology that is a crucial player in the remediation of harmful pollutants and industrial chemicals [1,2]. Further, nanotechnology is gaining attention in drug development due to its safety and other pharmacological properties [3]. Regarding environmentally friendly approaches, green chemistry-based nanomaterial creation employs naturally occurring materials, nontoxic chemicals, and biodegradable and biocompatible materials [4]. Plant-based phytochemicals or biomolecules have emerged as promising bio-reducing and capping/stabilizing agents for nanomaterial biofabrication due to their low cost, environmental friendliness, widespread availability, and long-term sustainability [5]. Several metals, including silver, cobalt, nickel, and palladium, have been examined in the form of NPs. They have distinctive properties that have pushed their use in various applications [6,7,8,9]. Metallic nanoparticles have a wide range of biological, catalytic, and electrochemical applications [10]. In medical science, metal- and metal oxide-supported nanomaterials have a significant therapeutic effect [11]. Due to their modest size-to-volume ratio and high thermal stability, metal nanoparticles are widely used in biological domains [12]. To date, many metal-based nanoparticles have been synthesized by different physical and chemical methods and have been found to have antibacterial properties [13]. Silver nanoparticles are the best among the various forms of metallic nanoparticles because of their broad antibacterial properties [14,15]. These silver nanoparticles cling to microbe cell walls and membranes, may reach the cytoplasm of the cell, and destroy cellular structures, which leads to the generation of reactive oxygen species and affects signal transduction pathways [16,17].

Similarly, Ni-NPs have piqued the interest of many researchers and have been used as an electro- and photo-catalyst, heat-exchanger, and biosensor. Furthermore, they have been increasingly popular in recent years because of their exceptional qualities, such as biocompatibility, reactivity, operational simplicity, affordability, abundance, anti-inflammatory actions, and environmental compatibility [18,19,20,21]. The magnetic, antibacterial, catalytic, and optical properties of bimetallic nanoparticles are excellently functionalized [22,23,24]. Surface plasmon resonance (SPR) spectrum shifts, a feature of its optical properties, have recently found use in plasmonic sensors [25,26]. One of the most promising nanomaterials is bimetallic nanoparticles, which exhibit a wide range of features due to the unique synergy generated when two different metals are incorporated into one particle. Antimicrobial agents and medication delivery systems can benefit from this phenomenon as imaging agents and imaging devices [27]. However, the global emergence of multi-drug resistance (MDR) among fungi, especially yeast, has presented a significant threat to the global public healthcare sector [28,29]. Due to the rising incidence of immune-compromising diseases and the unwarranted use of broad-spectrum antibiotics, fungal infections, notably with antifungal-resistant fungi, have complicated the therapeutic problems, requiring improved antifungal regimens [30]. Regardless of several developments in antifungal drug development, fungal infectious diseases such as *Candida* infections have high mortality rates [31]. Among existing *Candida* species, *C. albicans* is the utmost prevalent and problematic, and causes around 50% of candidiasis incidences. This is attributed to the striking ability of *C. albicans* to form a biofilm and to undergo morphological changes to survive in the host microenvironment [16,32]. Presently, fluconazole is the first-choice antifungal drug for combating various fungal diseases due to its high effectiveness, low price, and less toxic nature [33]. However, misuse of this drug and other members of the azole class of antifungal drugs has resulted in multi-drug resistant *C. albicans* [34,35]. Thus, an effective antifungal strategy is urgently needed [36]. The two important mechanisms causing drug resistance in *C. albicans* are mutations in the ergosterol biosynthesis pathway and upregulation of antifungal efflux [37,38,39]. Therefore, compounds that target these cellular machineries can provide us with options to combat drug resistance in *C. albicans*.

Metallic nanoparticles have unique physical and chemical properties, and therefore, they have broad applications for drug delivery, as antimicrobial agents, and in bio-detection [40,41]. Among the known nanoparticles, silver nanoparticles (Ag NPs) are well known for their antimicrobial properties, low toxicity, and high availability [42,43,44]. Silver nanoparticles and other metallic nanoparticles in combination with fluconazole (FLZ), nystatin (NYS) and ketoconazole (KTZ), have displayed increased effectiveness against *Candida* when compared to the drugs themselves [28,45]. Previously, Ag NPs exerted anti-*Candida* activity against *C. albicans* SC5314 by disrupting membrane integrity, mainly by unbalancing ergosterol content and by the composition of fatty acids [46]. Therefore, Ag NPs target those essential factors for drug resistance in *Candida* species [47]. Similarly, the antibacterial potency of Ag-Ni nanohybrids has been reported against *Streptococcus pyogenes* and *Escherichia coli* [48]. Hence, the present study aimed to study the anti-*Candida* effect of Ag-Ni NPs alone and in combination with FLZ and tried to investigate if Ag-Ni NPs treatment could lower the resistance of *C. albicans* to FLZ. Furthermore, the inhibitory actions of Ag-Ni NPs on biofilm production and efflux pumps of *C. albicans* were also studied. Therefore, this study may provide a substitute for traditional antifungal treatment and further confirms the antifungal potency of Ag-Ni NPs against *C. albicans*.

## 2. Materials and Methods

### 2.1. Materials

All chemicals, including silver nitrate (AgNO_3_) and nickel nitrate (Ni(NO_3_)_2_·6H_2_O), were of analytical grade and purchased from Sigma-Aldrich, St. Louis, MO, USA. Dried leaves from *Salvia officinalis* were purchased from the local marked in Jeddah, Saudi Arabia.

### 2.2. Preparation of Ag-Ni Nanoparticles

All solutions were prepared with deionized water, and analytical-grade chemicals were used without further purification. About 20 g of dried *Salvia officinalis* leaves were thoroughly washed, sliced to fine pieces, and ground with a pestle and mortar. The leaves were further immense in de-ionized water, heated at 60 °C for 2 h, and incubated at room temperature for one day. Later, the solution was filtered by vacuum suction filtration funnel using Whatman No.1 filter paper. The resulting extract was used as a capping agent and reducing agent to prepare Ag-Ni nanoparticles. Then, 2.0 × 10^−4^ equimolar solutions of silver nitrate and nickel nitrate were prepared, and 50 mL of each was mixed in a beaker under continuous stirring, during which no change in the color was observed. To this homogenous solution of metal precursors of silver and nickel, 60 mL of freshly prepared *Salvia officinalis* aqueous extract was added under continuous stirring at 60 °C for 20 min to acquire a clear, homogeneous solution. After that, the reaction mixture was continuously heated at 60 °C under constant vigorous magnetic stirring until a noticeable color change in the mixture was observed. This reaction mixture was further heated for 30 min at the same temperature, and the formation of the nanoparticles was observed by monitoring the absorbance using UV-visible spectrophotometer. Finally, the nanoparticles were isolated using the centrifugation process, which was carried out using a Thermo Fisher Scientific Centrifuge (Thermo Electron LED) at 10,000× *rpm* for 30 min. The procedure was repeated, and the finished product was re-dispersed in deionized water and centrifuged once more. The acquired Ag-Ni NPs were further dried in an oven at 120 °C for 12 h and finally calcined at 500 °C for 3 h before characterization.

### 2.3. Characterization of Ag-Ni Nanoparticles

Green chemistry-based Ag-Ni nanoparticles prepared by using *Salvia officinalis* aqueous were initially characterized using a double been UV-visible spectrophotometer (Shimadzu UV-1280) to confirm the biogenic reduction of Ag^+^ and Ni^2+^ to Ag-Ni bimetallic nanoparticles. Fourier transform infrared (FTIR) spectroscopy analysis was performed by using a Compact FT-IR Spectrometer ALPHA-II to assess the possible involvement of functional groups in the *Salvia officinalis* aqueous extract for the reduction and stabilization of the Ag-Ni nanoparticles. The crystal size and purity of the as-prepared Ag-Ni nanoparticles were determined using an X-ray diffractometer (Bruker D8 Advance) with a CuKα1 X-ray source (=1.54056) at 40 kV and a current of 40 mA in the range of 20 °C to 90 °C. The morphological examination of Ag-Ni nanoparticles was performed using a scanning electron microscope (SEM) (ZEISS model (Sigma VP. FESEM) equipped with an energy dispersive spectroscope (EDS). Energy-dispersive X-ray spectroscopy was used to investigate the elemental composition (EDX). Thermogravimetric analysis (TGA) was performed using a Perkin Elmer STA-8000 under a nitrogen environment at a heating rate of 10 °C per minute.

### 2.4. Microbiological Analysis

#### 2.4.1. Antifungal Susceptibility Profiling

The present study utilized *C. albicans* 5112 (FLZ-resistant strain) and *C. albicans* SC5314 (FLZ-sensitive strain) for investigating the antifungal potential of Ag-Ni NPs, by calculating minimum inhibitory concentration (MIC) and minimum fungicidal concentration (MFC) using the micro-broth dilution assay using standard method documented by CLSI (M27-A3) [49]. The initial concentration of Ag-Ni NPs was 400 µg/mL, and serial dilution was performed in a 96 well microtiter plate to achieve the desired concentrations (0.05–100 µg/mL) of nanoparticle, followed by the addition of 100 μL of *C. albicans* cells (5.0 × 10^6^ CFU/mL). FLZ (1024–0.5 µg/mL) was considered a positive control, whereas 1% DMSO was a negative control. After MIC determination, MFC was calculated by dispensing 10 µL from the wells without any turbidity on Sabouraud Dextrose Agar (SDA) growth medium, and plates were kept at 37 °C for 24 h. Following incubation, fungal growth for the dilution with >5 visible colonies was recorded as the MFC value.

#### 2.4.2. Cell Viability Assay

The susceptibility of *C. albicans* 5112 against Ag-Ni NPs was further confirmed by cell viability assay using Muse^TM^ Cell Analyzer (EMD Millipore, Darmstadt, Germany), following the protocol provided by the manufacturer. *Candida* cells were subjected to different Mic and 0.5× MICof test NPs for 4 h, followed by the Muse Count and Viability kit staining. Positive (heat killed yeast cells) and negative controls (unexposed cells) were also included in the experiment.

#### 2.4.3. Combinational Interaction of Ag-Ni NPs with Fluconazole

The anti-*Candida* potency of NPs in combination with FLZ against *C. albicans* 5112 was established according to the CLSI recommended standard protocol [50]. The test concentrations of Ag-Ni NPs used ranged from 0.05 to 100 µg/mL, and for FLZ were 1024–0.5 µg/mL. Thereafter, 50 µL of both nanoparticle and FLZ were dispensed into predefined wells, followed by the addition of 100 μL of *C. albicans* cells (5.0 × 10^6^ CFU/mL), and after that incubation at 37 °C for 24 h. Additionally, 1% DMSO was added as a negative control, along with growth and sterility controls for the experiment. The combination interaction was estimated by determining the fractional inhibitory concentration indexes (FICIs), which were calculated as follows:$$\mathrm{FICI}\;=\;\mathrm{FICa}\;+\;\mathrm{FICb}\;=\;\frac{\mathrm{MICa}\;\mathrm{in}\;\mathrm{combination}}{\mathrm{MICa}\;\mathrm{alone}}\;+\;\frac{\mathrm{MICb}\;\mathrm{in}\;\mathrm{combination}}{\mathrm{MICb}\;\mathrm{alone}}$$

MIC_a_ = MIC of the Ag-Ni NPs; MIC_b_ = MIC of FLZ. The FICI values were interpreted as: ≤0.5 = synergy, 0.5 and 1.0 = additive, 1.0 and 4.0 = indifferent, >4.0 = antagonistic.

#### 2.4.4. Effect on Morphological Transition

The impact of the Ag-Ni NPs on yeast to hyphal transition was estimated by adapting the procedure described previously [51]. Briefly, the cells were grown until late log phase was achieved. The synchronized yeast cell population was followed by washing with freshly prepared PBS and re-inoculation in fresh SDB at 37 °C for 48 h. To start the morphological transition, cells (100 μL) were allowed to grow in sterile SDB (20 mL) with fetal bovine serum (FBS; Sigma Aldrich Co., St. Louis, MO, USA; 10%). Ag-Ni NPs were added at 0.5 × MIC and MIC values, followed by incubation at 37 °C at 150 rpm for 4 h. Change in cell morphology was observed microscopically using a laser scanning confocal microscope. For comparison, a negative control of untreated *C. albicans* cells was used.

#### 2.4.5. Effect on Biofilm Development

The anti-biofilm activity of Ag-Ni NPs against *C. albicans* 5112 was studied as mentioned previously [52] by estimating the metabolic activity using semi-quantitative 2,3-bis(2-methoxy-4-nitro-5-sulfo-phenyl)-2H-tetrazolium-5-carboxanilide (XTT) reduction assay. Different concentrations ranging from 0.09 to 100 µg/mL of Ag-Ni NPs were tested. Then, 100 µL of *C. albicans* cells (5.0 × 10^6^ CFU/mL) was transferred to distinguished wells of microtiter plates and kept at 37 °C for 2 h followed by the removal of planktonic cells. All the wells were then gently washed with PBS. After that, different concentrations of Ag-Ni NPs (0.09–100 µg/mL) were transferred to the designated wells of the microtiter plate and incubated for 48 h. Metabolic activity of *C. albicans* adherent cells was estimated by XTT reduction assay, and readings were recorded at 490 nm using a SpectraMax iD3 multi-mode microplate reader.

To evaluate the fungal viability within the biofilms, CFU was calculated. Post-adhesion, the cells were treated with different concentrations of Ag-Ni NPs and incubated for 48 h at 37 °C. Later on, the attached biofilm was washed twice with PBS, scrapped out, and plated on SDA plates for determining the mean log CFU/mL values.

#### 2.4.6. Confocal Laser Scanning Microscopy (CLSM)

The anti-biofilm activity of Ag-Ni NPs was further confirmed by CLSM using the method reported by Sun et al. [53]. Different concentrations of Ag-Ni NPs were selected to be tested. For this test, fluorescent dyes FUN-1 (excitation wavelength = 543 nm and emission wavelength = 560 nm) and concanavalin A (ConA)-Alexa Fluor 488 conjugate (excitation wavelength = 488 nm and emission wavelength = 505 nm) were used. Slides were observed for biofilms under laser scanning confocal microscopy (Zeiss 780, Carl Zeiss, Jena, Germany).

#### 2.4.7. Efflux Assay

The efflux assay was performed by quantifying extracellular Rhodamine 6G (R6G) from *C. albicans* 5112 cells and *C. albicans* SC5314, as mentioned elsewhere [54]. To test the effect of the test nanoparticles, cells were subjected for 2 h to 0.5 × MIC and MIC of Ag-Ni NPs. All the *C. albicans* 5112 were de-energized using 2-deoxy-_D_-glucose (5.0 mM) and 2,4 dinitrophenol (5.0 mM) for 45 min, followed by 10 μM of R6G. The cells were then again incubated for 40 min at 37 °C, followed by washing and resuspension in glucose-free PBS. At 5 min for 1 h, an aliquot of 2 mL was taken and centrifuged at 3000× *rpm*, and OD was recorded for the supernatant at 527 nm. For energy-dependent efflux, 0.1 M glucose was added at the 25th minute of incubation. The OD was compared with the standard concentration curve of R6G to quantify the actual concentration of effluxed R6G.

#### 2.4.8. Intracellular R6G Accumulation Assay

As discussed earlier, an intracellular accumulation assay was performed with minor modifications [54]. For experimental purposes, *C. albicans* 5112 cells were treated with 0.5 × MIC and MIC values of Ag-Ni NPs for 2 h at 37 °C, followed by resuspension in 1 mL of PBS with 2% glucose and 4 µM R6G, and then further incubated for 30 min at 37 °C. After incubation, cells were pelleted and resuspended in cold PBS to prepare slides for fluorescence microscopy.

#### 2.4.9. Effect of Ag-Ni NPs on Membrane Integrity

A standard marker, propidium iodide (PI) was utilized to investigate the potential of Ag-Ni NPs on plasma membrane permeability in *C. albicans* following the protocol described previously [55], with modifications. Briefly, cells at a concentration of 5.0 × 10^6^ CFU/mL were subjected to 0.5 × MIC and MIC of Ag-Ni NPs for 4 h, followed by exposure to 30 µM of PI for 30 min in a dark room. After that, cells were harvested, washed with PBS, and used for fluorescence microscopy. In addition, cells exposed to 10 mM H_2_O_2_ were included as a positive control.

#### 2.4.10. Scanning Electron Microscopy

SEM was used to study the potency of Ag-Ni NPs on cellular architecture of *C. albicans* 5112. The yeast cells were treated with 0.5 × MIC and MIC of Ag-Ni NPs and incubated at 37 °C, 200 rpm for 48 h. Post incubation, aliquots of 100 µL were withdrawn and fixed with 2.5% glutaraldehyde for 2 h. Afterward, fixed cells were rewashed with PBS and were subjected to gradient dehydration with ethanol (40%, 10 min; 60%, 10 min; 80%, 10 min, and 100%, 20 min). Later, 20 μL aliquots of fixed and dehydrated cells were used to prepare slides and subjected to critical point drying, carbon-coated, and observation under the SEM (Zeiss Gemini 2 Crossbeam 540 FEG SEM, Oberkochen, Germany).

#### 2.4.11. Statistical Analysis

Data are represented as the averages of three independent experiments (mean ± SD) and were analyzed using Graph Pad Prism version 9.1.0 using Student’s *t*-tests (*p* value < 0.05).

## 3. Results and Discussion

### 3.1. Spectroscopic and Microscopic Analysis of Ag-Ni Bimetallic Nanoparticles

The present study reports the biogenic fabrication of Ag-Ni bimetallic nanoparticles using an aqueous extract of *Salvia officinalis* leaves. One of the most promising nanomaterials is the bimetallic nanoparticle. They can exhibit a wide range of features due to the unique synergy generated when two different metals are incorporated into one particle. Antibacterial agents and drug delivery systems can benefit from this phenomenon, enhancing or amplifying their usefulness as antimicrobial agents and imaging agents [56]. Biogenic fabrication of Ag-Ni nanoparticles was successfully performed via one-step seedless method using an aqueous extract of *Salvia officinalis* leaves through a biogenic reduction method. The synthesis process of Ag-Ni nanoparticles is depicted in the schematic diagram (Figure 1). The aqueous extract of *Salvia officinalis* is well known to be rich in numerous different types of phytochemicals, such as polyphenols, flavonoids, and alkaloids, which have a strong ability to reduce the metal ions to metallic nanoparticles through the stabilization and growth processes [20,57]. This makes *Salvia officinalis* extract a reducing agent and potential capping/stabilizing agent that makes the as-prepared nanoparticles highly biocompatible and stable [20,58]. After mixing the *Salvia officinalis* extract with silver nitrate, it went from light brown to dark brownish black. The change in color was the first visible sign of Ag-Ni nanoparticle formation under our optimal reaction conditions.

Using UV-visible spectroscopy, one may determine the stability and synthesis of metal nanoparticles in an aqueous solution. The biogenic synthesis of Ag-Ni nanoparticles was initially confirmed by UV-visible spectrophotometry by recording the absorbance of the *Salvia officinalis* aqueous extract, silver nanoparticles (Ag NPs), nickel nanoparticles (Ni NPs), and Ag-Ni nanoparticles in the wavelength range of 200–800 nm. The *Salvia officinalis* aqueous extract shows an absorption peak at 325 nm that could be due to the π-π transition [59], which signifies the presence of polyphenolic compounds present in the *Salvia officinalis* aqueous extract (Figure 2a). The absorption band of *Salvia officinalis* extract disappears in synthesized Ni NPs. A new absorption band appears at 290 nm, which shows that the phytochemicals present in *Salvia officinalis* extract are involved in the formation of Ni NPs (Figure 2a). The UV-vis spectrum in the wavelength range of 250–370 nm corresponds to the metallic nature of the Ni NPs [60]. The formation of Ag NPs using *Salvia officinalis* extract was monitored by observing the characteristic surface plasmon resonance (SPR) band using UV-visible spectroscopy. The SPR band observed at 435 nm confirms the successful formation of Ag NPs by utilizing *Salvia officinalis* aqueous extract, which worked as a reducing and stabilizing/capping agent at the same time (Figure 2a) [61]. The UV-visible absorption spectra of Ag-Ni nanoparticles prepared by the seedless one-step green synthesis approach are shown in Figure 2a. The absorption band of Ag-Ni nanoparticles is very broad and is shifted towards a higher wavelength, with the band being at 448 nm. The silver character is dominant over the nickel character, as the absorption for the Ag-Ni nanoparticles show a slight redshift. This is attributable to an increase in silver character, and an increase in particle size can also cause a redshift in the SPR. The aqueous extract of *Salvia officinalis* included phytomolecules that reduced Ag^+^ to Ag^0^ and Ni^2+^ to Ni^0^, leading to the creation of silver/nickel mono and silver/nickel bi-metallic nanoparticles.

FTIR spectroscopic analysis has been used to identify the functional groups that essentially take part in the reduction and capping/stabilization of the as-prepared Ag-Ni nanoparticles. The FTIR spectral comparison of *Salvia officinalis* aqueous extract with Ag-Ni nanoparticles shown in Figure 2b (black line) reveals noticeable peaks shifts at various wavenumbers, including 3340.8 cm^−1^ (O–H stretch), 2996.4 (C–H stretch), 1581.0 cm^−1^ (C=O/C–N/C=C stretching vibrations of functional groups), 1384.6 cm^−1^ (symmetric bending of methyl CH bonds), 991.1 cm^−1^, and 772.0 cm^−1^ (=C-H bending), which were consistent with the polyphenolic or flavonoid compounds present in the extract. The stretching vibration of the CH_2_ group is ascribed to the peaks at 2996.4 and 2915.4 cm^−1^, whereas the vibration peak at 1581.0 cm^−1^ suggests C=O stretching or C–N bending in the amide group. The FTIR spectral comparison of *Salvia officinalis* aqueous extract with Ag-Ni nanoparticles shown in Figure 2b (red line) reveals noticeable peaks shifts at various wavenumbers, including 3343.6 cm^−1^ (O–H stretch), 1577.9 cm^−1^ (C=O/C–N/C=C stretching vibrations of functional groups), 1313.4 cm^−1^ (symmetric bending of methyl CH bonds), 955.5 cm^−1^, and 762.1 cm^−1^ (=C-H bending), which are consistent with the polyphenolic or flavonoid compounds present in the extract. The FTIR spectral peaks of Ag-Ni NPs at 3334.6, 1577.9, 1313.4, 955.5, and 762.1 cm^−1^ indicate a shift in absorption bands of the phytochemical molecules from the aqueous extract of *Salvia officinalis* that perform the active roles of reducing and stabilizing agents. As a result, it was determined that glycosides, terpenoids, flavonoids, and alkaloids found in *Salvia officinalis* aqueous extract were responsible for the reduction and capping/stabilization of Ag-Ni nanoparticles. The current findings are consistent with those of other previously published studies on the green synthesis of Ag-Ni NPs [37,58,60,61].

The surface morphology and the elemental composition of prepared Ag-Ni nanoparticles were investigated by SEM and EDX analysis, respectively, as shown in Figure 3. The SEM analysis of *Salvia officinalis* extract-assisted Ag-Ni nanoparticles is displayed in Figure 3a. The structure and morphological behavior of the Ag-Ni bimetallic nanoparticles are supported by SEM analysis. From SEM analysis, it was observed that the bimetallic Ag-Ni nanoparticles have semi-spherical agglomerated clusters with an average size range from 31.84 to 47.85 nm. The co-existence of Ag-Ni NPs of smaller and larger sizes was due to the Ag NPs forming earlier as compared to the Ni NPs because of the large reduction potential of Ag (Ag^+^ + ē → Ag^0^ = +0.80 V) in comparison to Ni (Ni^2+^ + 2ē → Ni^0^ = −0.26 V), and in later stages of the reaction, the Ag NPs formation takes place on the surface. These findings suggest that nucleation of new nanoparticles and aggregation of bigger particles occurred simultaneously. EDX analysis was used to determine the elemental composition of Ag-Ni nanoparticles, and the findings are shown in Figure 3b. EDX data confirmed the existence of Ag and Ni components in as produced bimetallic nanoparticles. The Ag-Ni ratio was 52.36/47.64 in the samples, which is compatible with the Ag-Ni stoichiometric elemental ratio. To justify the cleanliness of metallic nanoparticles, the total metal content was relatively high. The existence of modest carbon and oxygen absorption intensity peaks at 0.25 and 0.5 keV, respectively, showed the presence of a sufficient surface coating of biomolecules as capping/stabilizing agents.

The crystallographic characterization and structural characterization of pure Ag NPs, Ni NPs, and bimetallic Ag-Ni nanoparticles were performed by X-ray diffraction (XRD) and by applications of the Rietveld-assisted Debye-Scherrer equation, as represented in Figure 3c. The obtained XRD patterns of Ag NPs and Ni NPs revealed that each diffraction peak is located at 2θ values observed at 38.443° (111), 44.621° (200), 64.754° (220), and 77.683° (311) for Ag NPs and 38.443° (111), 44.621° (200), 64.754° (220), and 77.683° (311) for Ni NPs. From the diffraction peaks of XRD pattern, it was observed that the diffraction peaks correspond to pure Ag (JCPDS card number 04-0783) and Ni (JCPDS card number 04-0850) crystal phases [62]. The diffraction planes for Ag NPs nanoparticles were found to follow a cubic structure, and Ni NPs were observed to be in a cubic crystal system and face-centered lattice structure. In addition, the typical diffraction pattern for Ag-Ni nanoparticles is represented in Figure 3c.

This indicates that the Ag-Ni NPs were crystalline with a face-centered cubic (fcc) structure. The diffraction pattern peaks of Ag-Ni NPs are about 38, 44, 64, and 77, which correspond to the (111), (200), (220), and (311) facets of Ag-Ni alloy, respectively. As Ni is a smaller atom in the Ag matrix and the introduction of Ni atoms does not create a significant change in the Ag lattice, the peaks for Ag-Ni NPs are similar to those for Ag alone. The high crystallinity of the *Salvia officinalis* extract may be related to the interactions of phytochemicals present in the extract. Ag-Ni NPs have an average size of 11.5 nm, according to a calculation of their size. The estimated crystalline size using the Scherrer formula and the measured crystalline size using a particle size analyzer were found to be in close agreement.

Thermogravimetric analysis and differential thermal analysis were used to investigate the thermal properties of Ag-Ni bimetallic nanoparticles in this study. Thermal analysis (TGA) and differential thermal analysis (DTA) spectra were acquired in the temperature range of room temperature to 800 °C at a heating rate of 10 °C/min utilizing a simultaneous thermal system (Shimadzu, DTG-60). Weight loss percent and derivative weight percent were plotted against temperature in the differential thermal analysis graph shown in Figure 3d, and the results were compared. The four stages of weight loss can be seen in the TGA graph, which are accompanied by three big endothermic peaks in the DTA graph. The vaporization of water molecules adsorbed on the surfaces of the Ag-Ni nanoparticles, which was observed between 45 °C and 115 °C, was connected with the first weight loss of 2.35 percent, which was recorded between 45 and 115 °C. Two additional weight losses were observed between 115 and 210 °C, and another between 210 and 325 °C, which were 10.86% and 9.42%, respectively, and could be associated with the decomposition of organic biomolecules present on the surface of Ag-Ni nanoparticles as they were being prepared. According to certain theories, this weight loss is caused by the degradation of phenolic acids, flavonoids, and carbohydrates, which were derived from the fruit extract and responsible for stabilizing Ag NPs. During the temperature range of 325 °C to 435 °C, a fourth weight loss of 7.73% was recorded, which could be attributed to the presence of thermal breakdown-resistant aromatic compounds on the surfaces of the Ag-Ni bimetallic nanoparticles. According to the results of the TGA-DTA analysis, the observed behavior in which a total weight loss of 30.36% occurred was due to the adsorbed water molecules and thermal decomposition of organic biomolecules/phytochemicals present on the Ag-Ni nanoparticles’ surfaces, which are responsible for the reduction of the Ag-Ni bimetallic nanoparticles and capping and stabilizing them. It is evident from the DTA profiles that the sample underwent complete thermal breakdown and crystallization simultaneously.

### 3.2. Antifungal Potential of Ag-Ni NPs

The Ag-Ni NPs showed potent antifungal activity against *C. albicans* FLZ-resistant and sensitive strains. The MIC value for FLZ-resistant strain was recorded as 1.56 µg/mL, whereas for the FLZ-sensitive strain, the value was 0.19 µg/mL. Additionally, the MFC value for the FLZ-resistant strain (3.12 µg/mL) was two-fold higher than its corresponding MIC value, whereas the MFC value for FLZ-sensitive strain (0.39 µg/mL) was only one-fold higher than its corresponding MIC value. Therefore, Ag-Ni NPs seems to have fungicidal activity against both *C. albicans* 5112 and *C. albicans* SC5314. The MIC value for FLZ against *C. albicans* 5112 was recorded as 64 µg/mL. With *C. albicans* being the most frequently reported pathogenic yeast in immunocompromised patients, we were looking for an efficient method for combating the resistant fungus. In this regard, combination therapy provides a convincing way to enhance the potency of available drugs, reduce toxicity, and most importantly, overcome antifungal drug resistance in *C. albicans* [63]. The Ag-Ni NPs showing fungicidal activity against the FLZ-resistant strain of *C. albicans* suggests that they could be a candidate for pharmaceutical and biomedical purposes. Although the antifungal activity of Ag-Ni NPs is not well reported in the literature, the antifungal activity of Ag NPs has been well documented in the literature. Different modes of action contribute to this nanoparticle’s fungicidal activity, for instance, generating reactive oxidation species (ROS), stopping ATP synthesis, altering cellular morphology, altering the membrane microenvironment, and altering cellular ultrastructure [64]. These attributes are very critical for developing resistance against commonly used antifungals and pathogenicity in *C. albicans*. Similarly to Ag NPs, Ni NPs also possess broad-spectrum antifungal activity [65], and therefore, we propose that the Ag-Ni NPs have the potential for targeting *C. albicans* efflux pumps and plasma membrane integrity, thereby resulting in cellular death. However, more investigations are needed to validate the present results. Data are represented as the averages of three independent experiments (mean ± SD) and were analyzed using Graph Pad Prism version 9.1.0 using Student’s *t*-tests (*p* value < 0.05).

An investigation was performed to determine the potencies of different concentrations of Ag-Ni NPs (0.5 MIC and MIC) on the cell survival profile of FLZ-resistant *Candida albicans*, and the findings are shown in Figure 4. Our results show that the Ag-Ni NPs at subinhibitory concentrations significantly reduced the viability of the cells as compared to the control cells in our study. When cells were subjected to 0.78 g/mL (0.5 MIC), we found 51.6 percent cell death; however, when the cells were exposed to 1.56 g/mL (1.5 MIC), we noticed a rise in the number of non-viable cells (71.9% at MIC value), and the result was statistically significant. For comparison, 1.1% and 99.1% cells were found dead in positive and negative controls, respectively.

The FLZ-resistant strain *C. albicans* 5112 was found to be sensitive to Ag-Ni NPs alone and in combination with FLZ. The MIC value of FLZ was considerably lowered, suggesting strong synergistic antifungal activity. After combinatorial treatment, the MIC value of Ag-Ni NPs was lowered from 1.56 to 0.19 μg/mL, whereas the MIC of FLZ was reduced from 64 to 0.12 μg/mL, and the FICI value was calculated as 0.31. Therefore, based on FICI values, we suggest that Ag-Ni NP exhibits an in vitro synergistic effect with FLZ against *C. albicans* 5112 (Appendix A). Besides the antifungal property of Ag-NPs, non-coated Ag NPs at higher concentration (5–10 μg/mL and above) have been reported to have cytotoxic activity in mammalian cells [66,67]. However, when used in combination with known drugs, the inhibitory concentrations of these nanoparticles were significantly reduced to the those which normally have less or no cytotoxicity. Therefore, combination therapy, while reducing the overuse of drugs, may give a suitable approach for combating resistance against commonly used antifungals. Furthermore, the synergistic effect of Ag NPs and azoles (a class of antifungals) against FLZ nonresponsive *C. albicans* has been discussed by researchers [6,68]. In the present work, we also observed that Ag-Ni NPs work synergistically with FLZ and made FLZ nonresponsive *C. albicans* cells susceptible to FLZ treatment.

The results depicted that the Ag-Ni NPs prevented yeast hyphae conversion in FLZ-resistant *C. albicans* 5112 (Figure 5a). On the contrary, the negative control (unexposed) cells presented very dense filamentous outgrowth in FBS. At the same time, the exposure to NPs resulted in a significant decline in hyphae formation in *C. albicans*. Exposure of *C. albicans* cells to 0.78 μg/mL (0.5 × MIC) of Ag-Ni NPs resulted in reduced hyphal growth compared to unexposed control cells. At 1.56 μg/mL (MIC) of NPs, we observed evident inhibition of filamentous growth, and well-defined yeast cells were observed under the microscope.

The anti-biofilm activity of Ag-Ni NPs against FLZ-resistant *C. albicans* 5112 was investigated. The results demonstrated that Ag-Ni NPs worked as an anti-biofilm agent at a concentration of 3.12 μg/mL, whereas the untreated control yeast cells could form a dense biofilm on the coverslip. Each biofilm was composed of a community of well-defined yeast and hyphal cells entrenched in extra polymeric substances (EPS). Biofilms among *Candida* species are considered a major virulence attribute and hinder the treatment of biofilm-associated infections in immunocompromised patients. Moreover, biofilms have been reported to show increased drug resistance and develop a varied and natural drug-tolerant environment. Drug-resistant *C. albicans* biofilms have been reported on several medical devices, and therefore are a challenge for treatment procedures [69,70]. In the current study, Ag-Ni NPs were found to be robust against FLZ-resistant *C. albicans* biofilms at a low concentration of 3.12 μg/mL. The average A_490_ value for healthy cells was recorded as 1.095, whereas the average A_490_ value for the cells exposed to 3.12 μg/mL of Ag-Ni NPs was recorded as 0.219, which was calculated as 80% inhibition of biofilm formation. Additionally, the Log_10_ CFU/mL for untreated control was 8.16, whereas the value was 6.72 after cells were exposed to 3.12 μg/mL of Ag-Ni NPs (Figure 5c). Despite there being decrease in cell count compared to untreated cells, the viability results confirmed that the effect was on biofilms and not due to cell death or growth inhibition. The present findings strongly advocate for Ag-Ni NPs having broad comprehensive prospects for early treatment or prevention of biofilm-related infections in immunocompromised individuals. Furthermore, antifungal and anti-biofilm activity of Ag NPs have been reported previously [71,72]. Therefore, our findings agree with the literature and strengthen our results for the anti-biofilm activity of Ag-Ni NPs against *C. albicans* biofilms.

The confocal microscopy further corroborated the anti-biofilm activity of Ag-Ni NPs. The biofilm formed in untreated FLZ-resistant *C. albicans* 5112 was a typical 3D structure, composed of hyphae. As Con A fluorescent dye binds with the biofilm matrix, the biofilm matrix was well-defined and dense, and it fluoresced bright green when exposed to ultraviolet light. However, biofilm formation was inhibited in Ag-Ni NP-treated *C. albicans* 5112. At a concentration of 0.78 μg/mL (0.5 × MIC) of Ag-Ni NPs, the biofilm lacked true hyphae and mainly comprised pseudohyphae and yeast cells. When 1.56 g/mL of Ag-Ni NPs was introduced to the biofilm, the biofilm architecture was significantly altered, consisting primarily of yeast cells and being devoid of either true hyphal or pseudohyphal structures. Furthermore, the cells’ vitality was changed, as evidenced by yellow-green fluorescence in the cells. The dye FUN-1 remained restricted to the cytoplasm of nonviable cells and fluoresced yellow-green, indicating the presence of metabolically dormant cells (Figure 5b).

The inhibitory potential of Ag-Ni NPs over MDR efflux transporters was investigated in FLZ-resistant *C. albicans* 5112. For comparison, fluconazole-susceptible *C. albicans* SC5314 was also tested under similar conditions. The exposure of Ag-Ni NPs showed a concentration-dependent modulation of drug efflux pumps in *C. albicans* 5112. At a concentration of 1.56 μg/mL (MIC) of Ag-Ni NPs, inhibition of energy-dependent R6G efflux was observed, whereas at a low concentration of 0.78 μg/mL (0.5 × MIC), there was no differences in the efflux of R6G, as observed in unexposed healthy *C. albicans* cells (Appendix A). In fluconazole-susceptible *C. albicans* SC5314, the efflux pumps were not active; therefore, there was no discharge of R6G from the cells in treated and untreated cells (Appendix A).

The role of membrane efflux transporters in antifungal drug resistance among *Candida* species is well documented. ATP-dependent efflux pumps (ABC superfamily) in FLZ-resistant *Candida* cells are reported to efflux R6G dye after their passive entry; therefore, in vitro efficiency of efflux pumps can be investigated by utilizing R6G (efflux substrate) [73,74]. Consequently, we investigated the impact of Ag-Ni NPs on drug efflux pumps. The addition of glucose in untreated *Candida* cells resulted in the efflux of the R6G dye because of membrane-bound drug transporter pumps. Previous studies have established that silver nanoparticles could decrease the abundance of Cdr1p and Cdr2p, resulting in lower efflux pump activity in *C. albicans* [75]. Similarly, the present Ag-Ni NPs were found to target this critical factor that plays a critical role in drug resistance in *C. albicans*.

The collection of R6G dye inside FLZ-resistant *Candida* cells further confirmed the alteration of drug efflux pumps due to the exposure to Ag-Ni NPs. In unexposed cells there was no accumulation of fluorescent dye, whereas the exposure of Ag Ni NPs resulted in high fluorescence signals, suggesting a high concentration of intracellular R6G (Figure 6). Hence, our findings align with earlier results [76], advocating a blockage of MDR efflux pumps’ activity by Ag-Ni NPs.

Membrane disruptive potential of Ag-Ni NPs against FLZ-resistant *C. albicans* 5112 was investigated by using PI. Exposure to Ag-Ni NPs resulted in disruption of the plasma membrane in *Candida* 5112 cells. PI diffusion was observed in the cells, and therefore, a higher number of PI positive yeast cells was observed under microscopic study (Figure 7). Furthermore, higher uptake of PI was observed with the growing concentration of Ag-Ni NPs; maximum uptake was observed at MIC values, followed by lower concentrations.

The microbial plasma membrane is essential for cells’ growth and survival because it acts as an obstruction to external environmental stresses. Therefore, compounds aiming at fungal plasma membrane could be considered possible lead compounds with increased efficacy for antifungal drug development. Furthermore, PI can enter disrupted cell membranes and is used as a marker nucleic acid. The process, namely, apoptosis or necrosis, causes damage to the plasma membrane and allows PI to enter the cells, resulting in red fluorescence [77]. Previously, researchers have reported the mechanism of action of AMPs, and their antimicrobial effect is attributed to their membrane permeabilization tendency [78,79,80]. Therefore, we speculate that the plasma membrane disruption after exposure to Ag-Ni NPs is their mode of action, which is accompanied by the generation of oxidative stress in FLZ-resistant *C. albicans* 5112.

The impact of Ag-Ni NPs on the cellular architecture of FLZ-resistant *C. albicans* 5112 was monitored by SEM, and the results are displayed in Figure 8. The untreated yeast cells possessed a uniform three-dimensional morphology and were healthy, having smooth and unbroken surfaces (Figure 8A). Yeast cells exposed to Ag-Ni NPs were found to be of varying sizes and irregular shapes, be compressed, have surface depressions, and present leakage of intracellular material; additionally, the cells seemed to be unhealthy (Figure 8B,C). The profound depressions observed on the yeast cells reflect the cytocidal effect of Ag-Ni NPs. Therefore, our results from PI uptake and SEM assay reflect that Ag-Ni NPs also disrupt the integrity of the fungal cell membrane, resulting in cell death.

## 4. Conclusions

In this study a facile, seedless, green synthesis method was used for the preparation of stable, semi-spherical, agglomerated Ag-Ni bimetallic nanoparticles with enhanced antifungal properties. Invasive *Candida* infections are becoming common in immunocompromised patients, and the presently used therapeutic regimen has limited efficacy. The as prepared Ag-Ni nanoparticles exhibit pronounced antifungal activity against *C. albicans*, an opportunist pathogenic yeast. Our results also established potent anti-biofilm activity of Ag-Ni NPs against an FLZ-resistant strain of *C. albicans*. Additionally, the nanoparticle was found to have efficient cell membrane disruption ability against *C. albicans*. Altogether, the results advocate for the potential of Ag-Ni NPs as a candidate for future antifungal drug discovery.

## Figures and Tables

**Figure 1 jof-08-00733-f001:**
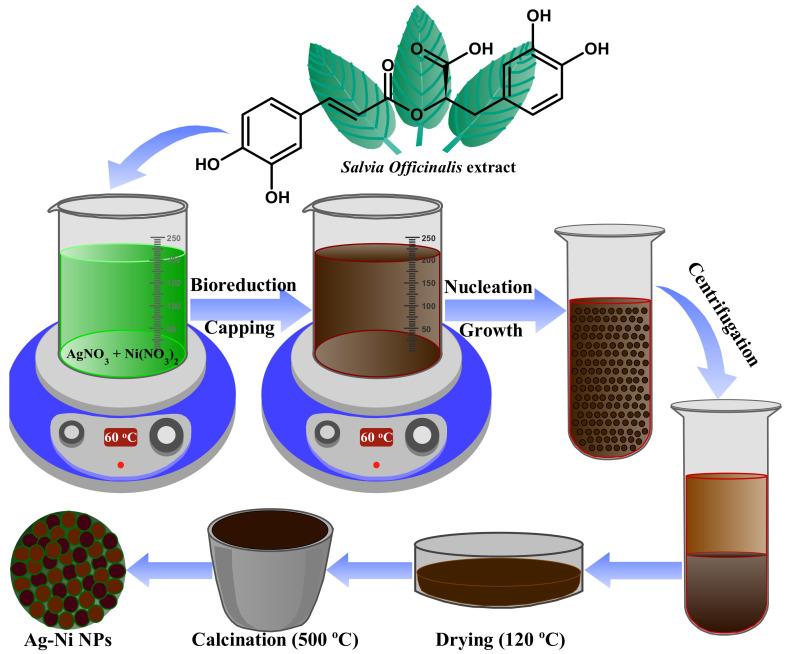
Schematic illustration of the formation of Ag-Ni NPs using *Salvia officinalis* aqueous extract as a stabilizing and reducing agent.

**Figure 2 jof-08-00733-f002:**
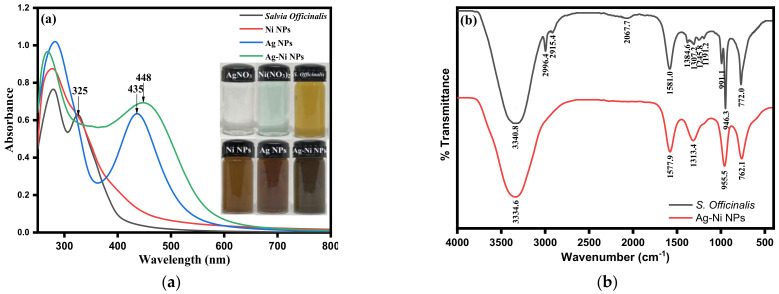
(**a**) UV–vis spectra of *Salvia officinalis* Ag NPs, Ni NPs and Ag-Ni NPs at 30 °C (inset optical images), and (**b**) Fourier transform infrared (FTIR) spectra of *Salvia officinalis* extract and Ag-Ni NPs.

**Figure 3 jof-08-00733-f003:**
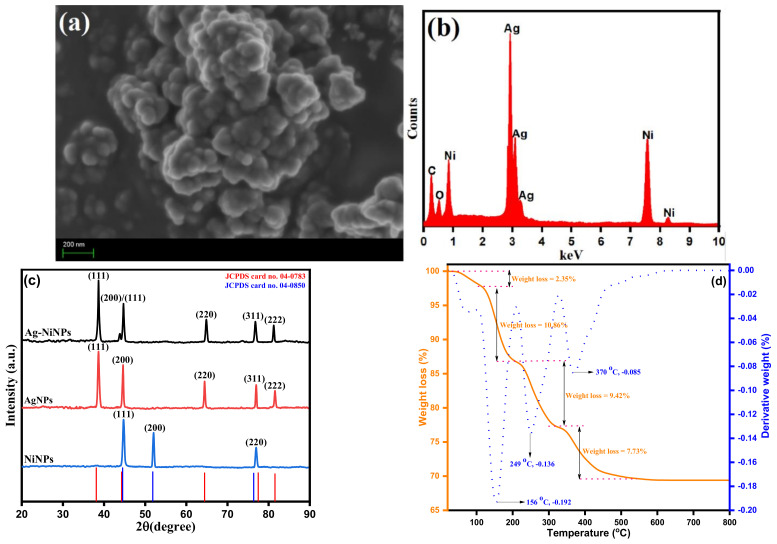
(**a**) Scanning electron microscopy (SEM) image, (**b**) energy-dispersive X-ray spectroscopy (EDX), (**c**) X-ray diffraction (XRD) patterns and (**d**) thermal gravimetric analysis-differential thermal analysis (TGA-DTA) curves of bimetallic Ag-Ni nanoparticles.

**Figure 4 jof-08-00733-f004:**
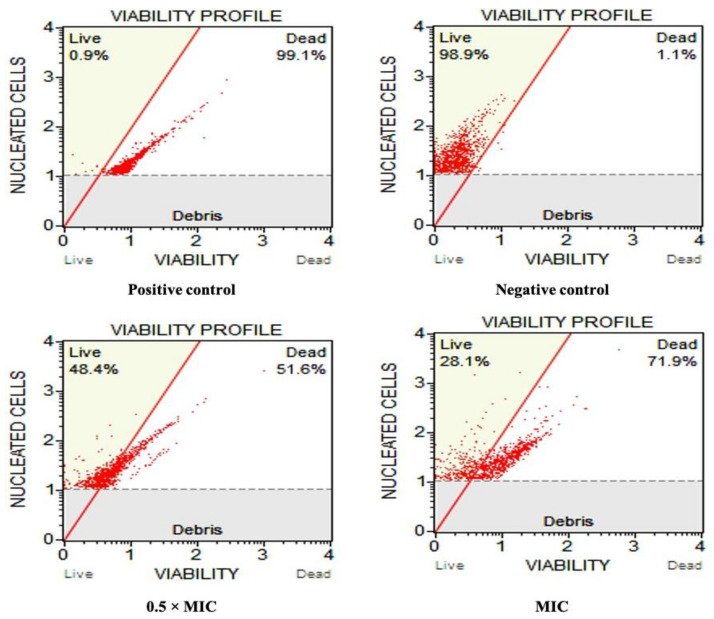
Viability profile of *Candida albicans* 5112 untreated cells (negative control), heat killed cells (positive control) and cells treated with 0.78 µg/mL (0.5 × MIC) and 1.56 µg/mL (MIC) of the bimetallic Ag-Ni NPs.

**Figure 5 jof-08-00733-f005:**
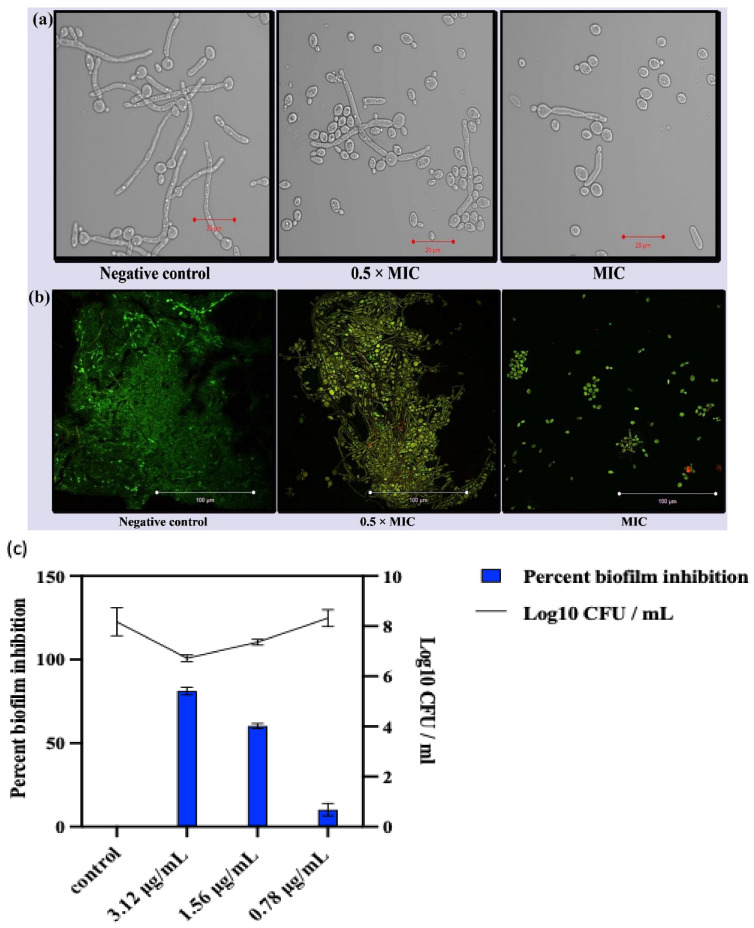
(**a**) Inhibition of morphogenesis in *C. albicans* 5112. Hyphae formation was monitored at different concentrations of Ag-Ni NPs (0.78 μg/mL, 0.5 × MIC; 1.56 μg/mL MIC). Untreated *C. albicans* 5112 was used as a negative control. (**b**) Anti-biofilm activity of Ag-Ni NPs. The picture shows the effect of Ag-Ni NPs on biofilm formation in *C. albicans* 5112. The biofilm matrix fluoresces green (Con A), whereas metabolically inactive cells fluoresce yellow-green (FUN-1). (**c**) Percent biofilm inhibition and cell viability within the biofilms with regard to different concentrations of Ag-Ni NPs.

**Figure 6 jof-08-00733-f006:**
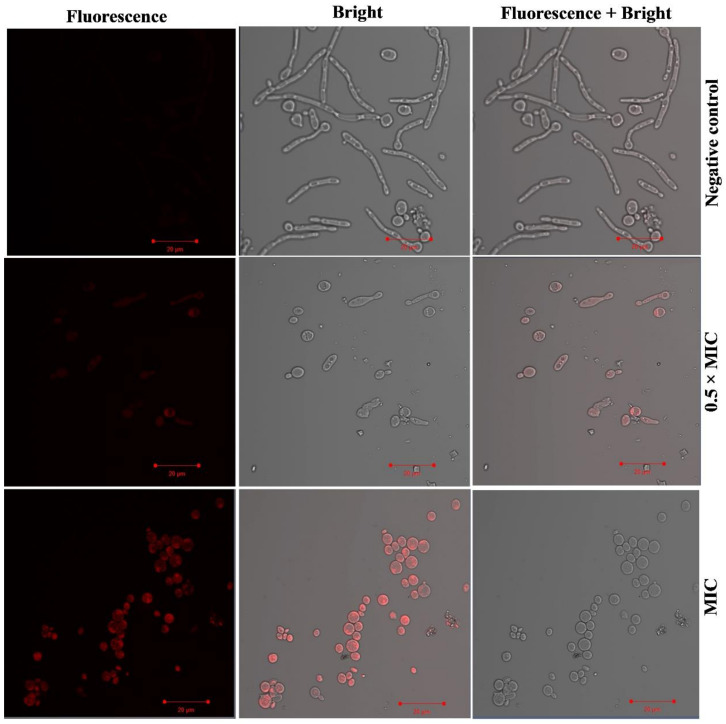
Intracellular accumulation of R6G in FLZ-resistant *Candida* cells. Fluorescence microscopy of R6G-stained FLZ-resistant *Candida* 5112. In the negative control, there was no accumulation of the dye after addition of glucose, whereas exposure to 0.5 × MIC (0.78 μg/mL) and MIC (1.56 μg/mL) of Ag-Ni NPs resulted in accumulation of R6G inside the *C. albicans* 5112 cells.

**Figure 7 jof-08-00733-f007:**
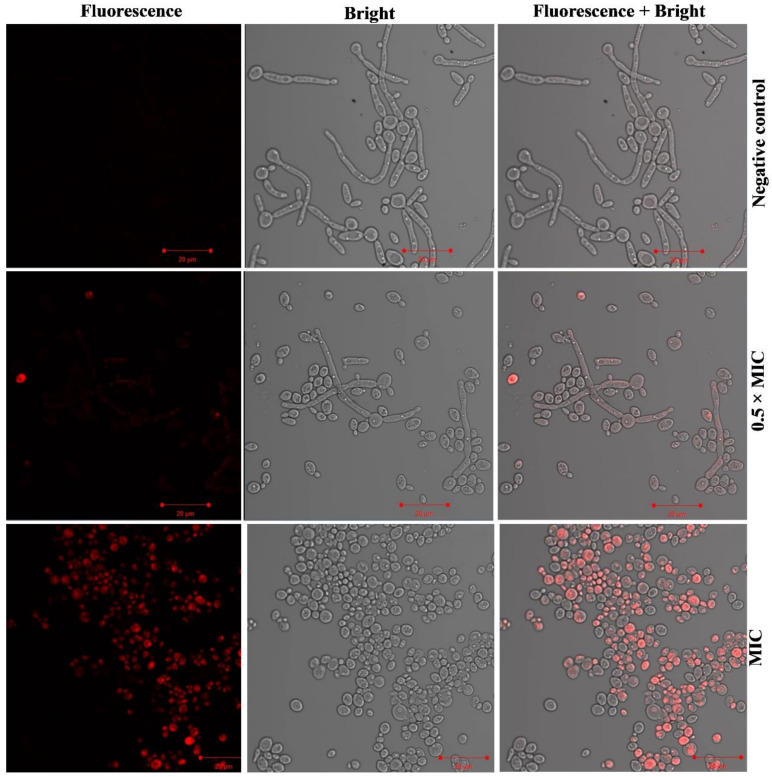
Incorporation of PI by FLZ-resistant *C. albicans* 5112. The figure shows the membrane disruption ability of Ag-Ni NPs against *C. albicans* cells. The negative control had cells with intact cell membrane.

**Figure 8 jof-08-00733-f008:**
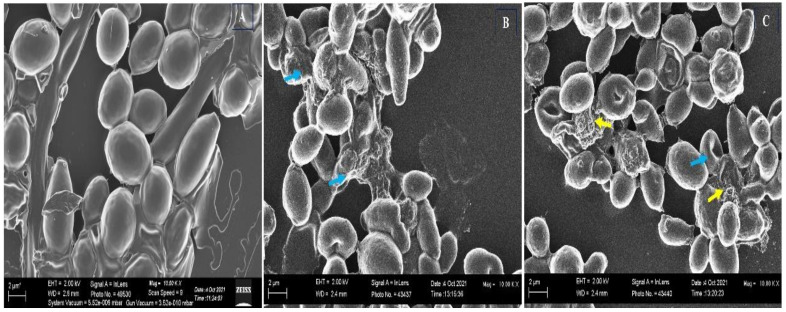
SEM images of FLZ-resistant *C. albicans* 5112. (**A**) Untreated control; yeast cells exposed to Ag-Ni NPs at different contractions: (**B**) 0.5 × MIC (0.78 μg/mL) and (**C**) MIC (1.56 μg/mL). Blue arrows show wrinkles, rupture, and distortion of *C. albicans* surfaces, whereas yellow arrows show discharge of intracellular material.

## Data Availability

The data available are provided in this manuscript.

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
