# Peer review of "Combination Effect of Novel Bimetallic Ag-Ni Nanoparticles with Fluconazole against Candida albicans"

_jof, 2022, doi:10.3390/jof8070733_

Round 1
Reviewer 1 Report
Thank you for giving me a chance to review this paper, which was very interesting and helpful.
The authors have compiled a descriptive study that attempts to propose an environmentally friendly and economical method for the biofabrication of silver nanoparticles using aqueous extract of Salvia officinalis. The manuscript appears to be an innovative approach as it may present an alternative to the traditional treatment for MDR Candida albicans infections. The authors have brought in significant investigation, but some major and minor issues need to be addressed. In addition, an extensive review for correction of typo, details in scientific terms, reference list and citations and figures needs to be carried out carefully, as the article is still deficient in this regard.
Major Issues
- The authors carry out tests to assess the antifungal potential of Ag-Ni NPs according to M27-A3. However, the document is not included in the reference list (line 397). The results presented Ag-Ni NPs antifungal activity against C. albicans FLZ resistant. However, the manuscript does not present tests using Candida albicans strains not resistant to fluconazole. I think it is necessary to use a non-resistant reference to support the data presented. By the way, manuscript introtuction fail off in describe a revision about the use of Ag-Ni NPs against C. albicans. Moreover, authors should pay attention to the text between lines 419 and 427, which is very confusing. These results is using Ag-Ni NPs alone or in combination with FLZ?
- Line 438-445
Please, explore this “Besides the antifungal property of AgNPs, the higher concentration of these nanoparticles has been shown to have cytotoxic activity in mammalian cells [64,65].” What is higher? Moreover, authors should pay attention to the text, which is very confusing and withou focus.
- The authors should note how many times the experiments have been repeated. If this is a one time-experiment, this would be a flaw in the design. Showing reproducibility would greatly enhance the potential benefits from this study. Please inform this in all the experiments performed
Minor issues:
- Line 22:
Please, use “Salvia officinales”
- Line 29:
Please, specify “MDR” in Abstract section C. albicans
- Line 68
Please, correct “characteristicare”
- Line 78
“fungus,ave”
-Line 85
Is Fluconazole the first choice antifungal drug for C. albicans? Please, clarify this issue.
-Line 99
Please, confirm that (45, 46) references are properly cited: “Ag NPs in combination with fluconazole (FLZ), nystatin (NYS) and ketoconazole 97 (KTZ) has displayed increased effectiveness against C. albicans when compared to drugs”
Lines 101-102
-Please, add reference for “Therefore, Ag NPs target those essential factors for drug 101 resistance in Candida species.”
Line 106
Please, change “provide” for “may provide a novel approach to treat Candida albicans infections ...”
Line 112
Please, add dot (.)
- Line 121
Please, correct “50mL”
- Lines 123-129
Please, correct the font style
- Line 162
Please, change “aliquoting”
- Line 162
Please, add “DAS” componentes
- Line 174
Revise reference for CLSI - standard method documented (48??)
-Line 209
Please, add “CLSM” in the title
- Line 228
Please, correct “performedith”
- Line 235
Please, add “(PI)” after “propidium iodide”
- The figures can be improved and revised to be more informative and the legends should be improved for clarity and understandability.
Please, correct Salvia Officinales in Figure 1 and lines 276 and 279, too. In this sense, all manuscript must be properly revised.
Nanoalloy?
What is the correct JPF form to cite the figures? Please, use only one and correct form to this.
(Fig. 2a). Line- 279
(Fig. 2(a)). Line 282
(Fig. 2(a)) Line 287
-Line 409
Please, use 'Reactive oxygen species' (ROS)
-Line 442
Please, correct “t synergistic effect”
- Line 458
Please, correct “floureses”
-Line 475
Please, correct “ aifungal ant-biofilm activity”
Author Response
Comments and Suggestions for Authors
Thank you for giving me a chance to review this paper, which was very interesting and helpful.
The authors have compiled a descriptive study that attempts to propose an environmentally friendly and economical method for the biofabrication of silver nanoparticles using aqueous extract of Salvia officinalis. The manuscript appears to be an innovative approach as it may present an alternative to the traditional treatment for MDR Candida albicans infections. The authors have brought in significant investigation, but some major and minor issues need to be addressed. In addition, an extensive review for correction of typo, details in scientific terms, reference list and citations and figures needs to be carried out carefully, as the article is still deficient in this regard.
Major Issues
# The authors carry out tests to assess the antifungal potential of Ag-Ni NPs according to M27-A3. However, the document is not included in the reference list (line 397). The results presented Ag-Ni NPs antifungal activity against C. albicans FLZ resistant. However, the manuscript does not present tests using Candida albicans strains not resistant to fluconazole. I think it is necessary to use a non-resistant reference to support the data presented. By the way, manuscript introtuction fail off in describe a revision about the use of Ag-Ni NPs against C. albicans. Moreover, authors should pay attention to the text between lines 419 and 427, which is very confusing. These results is using Ag-Ni NPs alone or in combination with FLZ?
Response: Authors highly appreciate the learned reviewers’ comments and suggestion that help use to improve our manuscript.
Below are point wise answers to your query.
- The reference for CLSI, M27-A3 has been added to the revised manuscript.
- In this study, only fluconazole resistant strains were used as the thesis of the paper is to reverse the drug resistance and therefore it will not make sense to use drug susceptible isolates. Furthermore, combination of drugs is required to treat resistant pathogens and therefore all the focus was on the resistant pathogens.
- Ag-Ni nanohybrids are not well explored for their anti-Candida activity; however, their antibacterial potential is known. Therefore, the gaps in the introduction section have been filled by mentioning the antibacterial activity of Ag-Ni NPs.
- The text between line 419-427 has been modified for better understanding of readers.
- For clarification, inhibitory effects of Ag-Ni NPs on efflux pumps and plasma membrane integrity in albicans were determined alone whereas, a separate experiment “Combinational interaction of Ag-Ni NPs with fluconazole” was designed to understand the interaction of Ag-Ni with FLZ. It is clearly mentioned in the experimental section where combinations were used.
# Line 438-445: Please, explore this “Besides the antifungal property of AgNPs, the higher concentration of these nanoparticles has been shown to have cytotoxic activity in mammalian cells [64,65].” What is higher? Moreover, authors should pay attention to the text, which is very confusing and without focus.
Response: Studies have suggested that non-coated Ag NPs are cytotoxic in mammalian cells at a concentration above 5-10 μg/mL. This information has been updated in the revised manuscript. The whole statement has been re-written for better understanding and to avoid any confusion (lines 527-535).
# The authors should note how many times the experiments have been repeated. If this is a one time-experiment, this would be a flaw in the design. Showing reproducibility would greatly enhance the potential benefits from this study. Please inform this in all the experiments performed
Response: Thank you for raising this point. For ensuring reproducibility of results all the experiments were performed in triplicate and at least three times. Also, the revised manuscript has been updated with statistical analysis section.
# Minor issues:
- Line 22: Please, use “Salvia officinales”
Response: Correction has been done.
- Line 29: Please, specify “MDR” in Abstract section C. albicans
Response: The whole sentence has been removed from the abstract based on other reviewer’s comment.
- Line 68: Please, correct “characteristicare”
Response: Correction has been done.
- Line 78: “fungus,ave”
Response: Correction has been done.
# Line 85: Is Fluconazole the first choice antifungal drug for C. albicans? Please, clarify this issue.
Response: Fluconazole is still the most widely used drug, and amphotericin B deoxycholate is also frequently used for broader coverage of candidemia. Although the 2009 guidelines for candidemia treatment by the Infectious Diseases Society of America recommend echinocandins as the first-line antifungal agents in the setting of neutropenia or moderate to severe illness, these drugs are not widely used worldwide as first-line agents due to their relatively high costs, especially in resource-limited countries. The sentence has been referenced for easy citation in the revised manuscript.
# Line 99: Please, confirm that (45, 46) references are properly cited: “Ag NPs in combination with fluconazole (FLZ), nystatin (NYS) and ketoconazole 97 (KTZ) has displayed increased effectiveness against C. albicans when compared to drugs”
Response: The statement and the citations has been updated.
# Lines 101-102: Please, add reference for “Therefore, Ag NPs target those essential factors for drug 101 resistance in Candida species.”
Response: An appropriate reference has been added to the text in the revised manuscript.
# Line 106: Please, change “provide” for “may provide a novel approach to treat Candida albicans infections ...”
Response: Correction has been done.
# Line 112: Please, add dot (.)
Response: Correction has been done.
# Line 121: Please, correct “50mL”
Response: Correction has been done.
# Lines 123-129: Please, correct the font style
Response: Font style has been changed to match the format of the paper.
# Line 162: Please, change “aliquoting”
Response: Aliquoting has been replaced by dispensing in the revised manuscript.
# Line 162: Please, add “DAS” components
Response: Full form of SDA has been provided in the revised manuscript.
# Line 174: Revise reference for CLSI - standard method documented (48??)
Response: Correct reference of CLSI has been added to the revised reference list.
# Line 209: Please, add “CLSM” in the title
Response: CLSM has been added to the title as suggested
# Line 228: Please, correct “performedith”
Response: Correction has been done.
# Line 235: Please, add “(PI)” after “propidium iodide”
Response: PI has been added after propidium iodide in the revised manuscript.
# The figures can be improved and revised to be more informative, and the legends should be improved for clarity and understandability.
Response: Information added as suggested.
Please, correct Salvia Officinales in Figure 1 and lines 276 and 279, too. In this sense, all manuscript must be properly revised.
Response: All changes made throughout the manuscript according to the suggestions of learned reviewer.
Nanoalloy?
Response: Corrected.
What is the correct JPF form to cite the figures? Please, use only one and correct form to this.
(Fig. 2a). Line- 279
(Fig. 2(a)). Line 282
(Fig. 2(a)) Line 287
Response: Same format has been used for figure representation throughout the manuscript.
#Line 409: Please, use 'Reactive oxygen species' (ROS)
Response: Full form has now been provided in the revised manuscript.
# Line 442: Please, correct “t synergistic effect”
Response: Correction has been done as suggested.
# Line 458: Please, correct “floureses”
Response: Correction has been done as suggested.
# Line 475: Please, correct “aifungal ant-biofilm activity”
Response: Correction has been done as suggested.
Reviewer 2 Report
This manuscript, entitled “Combination Interaction of Novel Bimetallic Ag-Ni Nanoparticle with Fluconazole to Reverse Drug Resistance in Candida albicans”, had described the preparation, characterization, and antifungal activity of Ag-Ni bimetallic nanoparticles (Ag-Ni NPs). This study is of interest, however, due to some significant drawbacks, my suggestion is major revision.
First and most importantly, the title is confusing. What does “Reverse Drug Resistance in Candida albicans” mean? How could the drug resistance be reversed? Does the Ag-Ni NPs induce the deletion or mutation of drug resistance genes? Such information is not included in the manuscript. If it only inhibits the drug (Fluconazole) resistance level of Candida albicans at a certain point, “reverse” is not appropriate. Please revise the title accordingly.
Secondly, the antibiofilm effect of Ag-Ni NPs on C. albicans has been tested by XTT assay. However, the data is not shown. It would be more convincing to show the change of OD490 values. Also, CV assay and CFU counting is recommended to be included to evaluate antibiofilm effect.
Thirdly, only phenotypical antifungal activity of Ag-Ni NPs is tested, which lacking depth. I would be interested to see the changes in biofilm formation (ALS1, ALS3, etc), drug resistance (ERG3, ERG11, etc.), hyphal formation (EFG1, CPH1, etc), and pathogenesis (ECE1, etc.) related genes in C. albicans upon Ag-Ni NPs treatment. Recently, an ECE1 variant had been identified in a clinical isolate 529L, which lower pathogenicity than wild type strain SC5314, contributed by sequence difference. It would be significance if Ag-Ni NPs would induce the sequence variation in ECE1 and change pathogenicity.
Fourthly, the structure of the abstract should be revised. It’s hard to understand at the current version. Background, methods, results, and conclusion should be included in abstract in a structured way. Overall, the formatting in the whole text should be carefully revised.
Author Response
Comments and Suggestions for Authors
This manuscript, entitled “Combination Interaction of Novel Bimetallic Ag-Ni Nanoparticle with Fluconazole to Reverse Drug Resistance in Candida albicans”, had described the preparation, characterization, and antifungal activity of Ag-Ni bimetallic nanoparticles (Ag-Ni NPs). This study is of interest, however, due to some significant drawbacks, my suggestion is major revision.
# First and most importantly, the title is confusing. What does “Reverse Drug Resistance in Candida albicans” mean? How could the drug resistance be reversed? Does the Ag-Ni NPs induce the deletion or mutation of drug resistance genes? Such information is not included in the manuscript. If it only inhibits the drug (Fluconazole) resistance level of Candida albicans at a certain point, “reverse” is not appropriate. Please revise the title accordingly.
Response: We are thankful to the learned reviewer for thoroughly reviewing our manuscript that help use to improve the quality of the article.
Authors agree with the reviewer’s comment and therefore changed the title which is straight reflecting the outcome of this study.
# Secondly, the antibiofilm effect of Ag-Ni NPs on C. albicans has been tested by XTT assay. However, the data is not shown. It would be more convincing to show the change of OD490 values. Also, CV assay and CFU counting is recommended to be included to evaluate antibiofilm effect.
Response: As suggested, the A490 values for both untreated control and treated biofilms along with the percentage biofilm inhibition value has been added to the result section.
Authors agree that CV and CFU counting is also recommended for evaluation of antibiofilm effect however, XTT is very sensitive test and combines the benefits of CV and CFU together, it measures the metabolic activity of cells so we can check the cell viability within the biofilm which gave us a better insight over antibiofilm activity of Ag-Ni NPs.
# Thirdly, only phenotypical antifungal activity of Ag-Ni NPs is tested, which lacking depth. I would be interested to see the changes in biofilm formation (ALS1, ALS3, etc), drug resistance (ERG3, ERG11, etc.), hyphal formation (EFG1, CPH1, etc), and pathogenesis (ECE1, etc.) related genes in C. albicans upon Ag-Ni NPs treatment. Recently, an ECE1 variant had been identified in a clinical isolate 529L, which lower pathogenicity than wild type strain SC5314, contributed by sequence difference. It would be significance if Ag-Ni NPs would induce the sequence variation in ECE1 and change pathogenicity.
Response: The authors agree with the reviewers’ comment that the present work demonstrates the antifungal potency of Ag-Ni NPs as well as its possible mode of action against FLZ resistant strain of C. albicans. It would have been interesting to see the impact of Ag-Ni on genetic level however, due to the limited resources and non-availability of the mutant strains we will not be able to include such studies in this project. We will keep the reviewer’s comment as a suggestion for future research projects.
# Fourthly, the structure of the abstract should be revised. It’s hard to understand at the current version. Background, methods, results, and conclusion should be included in abstract in a structured way. Overall, the formatting in the whole text should be carefully revised.
Response: As suggested by the reviewer, the abstract has been overall modified in the revised manuscript and it follows the style of structured abstract but without headings (formatted according to JoF). Furthermore, whole manuscript has been edited carefully to avoid any type scientific and language mistakes.
Round 2
Reviewer 1 Report
Considering the responses made to my comments and the content of the revised manuscript, the authors did a good job of addressing the most important points I raised in my original review. The title change was also quite adequate and presents the result of the work.
However, I respectfully disagree with the authors’ statement that - only fluconazole resistant strains were used as the thesis of the paper is to reverse the drug resistance and therefore it will not make sense to use drug susceptible isolates.
I still think it's important to test the compounds on non-resistant strains of Candida. I understand the focus was on resistant pathogens, but non-resistant strains would be adequate controls. The authors describe that “Ag-Ni nanohybrids are not well explored for their anti-Candida activity; however, its antibacterial potential is known” (review 1). The use of non-resistant Candida may increase knowledge about the effects of this new antifungal potential - Ag-Ni NPs and their activity alone or in combination with fluconzaol on resistant or non-resistant strains. I see no point in testing a new drug and not evaluating its effects on the reference strain.
# Susceptible Candida strain should be used as a control in some experiments (MIC, MFC,)
https://www.frontiersin.org/articles/10.3389/fmicb.2019.01021/full
# The authors showed an important action of the compound on resistant strains, but at the conclusion of the manuscript they sustain the statement that it was removed from the title - the compound leads to a reversion in resistance - which is questionable, considering the results presented. I suggest to present as a finding only the fungicidal effect in strains resistant to fluconazole.
Author Response
Considering the responses made to my comments and the content of the revised manuscript, the authors did a good job of addressing the most important points I raised in my original review. The title change was also quite adequate and presents the result of the work.
However, I respectfully disagree with the authors’ statement that - only fluconazole resistant strains were used as the thesis of the paper is to reverse the drug resistance and therefore it will not make sense to use drug susceptible isolates.
I still think it's important to test the compounds on non-resistant strains of Candida. I understand the focus was on resistant pathogens, but non-resistant strains would be adequate controls. The authors describe that “Ag-Ni nanohybrids are not well explored for their anti-Candida activity; however, its antibacterial potential is known” (review 1). The use of non-resistant Candida may increase knowledge about the effects of this new antifungal potential - Ag-Ni NPs and their activity alone or in combination with fluconzaol on resistant or non-resistant strains. I see no point in testing a new drug and not evaluating its effects on the reference strain.
# Comment: Susceptible Candida strain should be used as a control in some experiments (MIC, MFC,)
https://www.frontiersin.org/articles/10.3389/fmicb.2019.01021/full
# Response: As suggested by the reviewer, the MIC and MFC results for Fluconazole sensitive strain (C. albicans SC5314) has been added to the method and results section. Also, R6G efflux assay with sensitive FLZ strain was performed and results have been added to the revised manuscript.
# Comment: The authors showed an important action of the compound on resistant strains, but at the conclusion of the manuscript they sustain the statement that it was removed from the title - the compound leads to a reversion in resistance - which is questionable, considering the results presented. I suggest to present as a finding only the fungicidal effect in strains resistant to fluconazole.
# Response: The authors agree with the reviewer’s comment and therefore, the conclusion and other sections of the study have been modified by removing the term reversion and to present the results of this study.
Reviewer 2 Report
The manuscript has been slightly improved in the revised version, with title revised and OD490 value from XTT assay added. However, other comments were not addressed.
Firstly, concerning the strategies to assess antibiofilm effect. The authors claim “XTT is very sensitive test and combines the benefits of CV and CFU together”. I suggest the authors to check the principle of each assay and consider the difference among CV, CFU and XTT assays. They are used to measure different aspect of biofilm formation activity, biomass, culturability, and viability, respectively. There is no way XTT combines the benefits of CV and CFU.
Secondly, considering the gene expression level difference in biofilm formation, drug resistance, hyphal formation and pathogenesis related genes, mutant strains are not required. A simple RT-PCR is available to examine the changes in gene expression levels.
Author Response
The manuscript has been slightly improved in the revised version, with title revised and OD490 value from XTT assay added. However, other comments were not addressed.
Firstly, concerning the strategies to assess antibiofilm effect. The authors claim “XTT is very sensitive test and combines the benefits of CV and CFU together”. I suggest the authors to check the principle of each assay and consider the difference among CV, CFU and XTT assays. They are used to measure different aspect of biofilm formation activity, biomass, culturability, and viability, respectively. There is no way XTT combines the benefits of CV and CFU.
Response: Authors agree with the reviewer’s comment that the different assays have different principles. The anti-biofilm activity in this study has been confirmed by XTT and CLSM and therefore including CV will be an additional assay showing the same results. For CFU, we repeated the XTT assay to confirm the CFU and viability of cells in biofilms and the results have been briefly included in results section and Fig 5. These results advocate that the effect is on biofilms and not due to the cell death or growth inhibition.
Secondly, considering the gene expression level difference in biofilm formation, drug resistance, hyphal formation and pathogenesis related genes, mutant strains are not required. A simple RT-PCR is available to examine the changes in gene expression levels.
Response: Author’s agree with the reviewer’s comment and are aware that a Real Time PCR assay can be used to check the different levels of expression for the mentioned genes. It would have been ideal to include this study; however as mentioned due to resource and time limitations we will not be able to include this study in this paper. We will kindly keep this suggestion for the future research.